# A New Induction Method for the Controlled Differentiation of Human-Induced Pluripotent Stem Cells Using Frozen Sections

**DOI:** 10.3390/cells10112827

**Published:** 2021-10-21

**Authors:** Susumu Tadokoro, Reiko Tokuyama-Toda, Seiko Tatehara, Shinji Ide, Hirochika Umeki, Keiko Miyoshi, Takafumi Noma, Kazuhito Satomura

**Affiliations:** 1Department of Oral Medicine and Stomatology, Tsurumi University School of Dental Medicine, 2-1-3, Tsurumi, Tsurumi-ku, Yokohama 230-8501, Japan; tadokoro-susumu@tsurumi-u.ac.jp (S.T.); tokuyama-r@tsurumi-u.ac.jp (R.T.-T.); tatehara-s@tsurumi-u.ac.jp (S.T.); ide-shinji@tsurumi-u.ac.jp (S.I.); umeki-h@tsurumi-u.ac.jp (H.U.); 2Department of Oral Bioscience, Tokushima University Graduate School of Biomedical Sciences, 3-18-15, Kuramoto, Tokushima 770-8504, Japan; miyoshi@tokushima-u.ac.jp; 3Department of Nutrition and Health Promotion, Hiroshima Jogakuin University, 4-13-1, Ushitahigashi, Higashiku, Hiroshima 732-0063, Japan; noma@gaines.hju.ac.jp

**Keywords:** iPSC, differentiation, frozen section

## Abstract

Considering that every tissue/organ has the most suitable microenvironment for its functional cells, controlling induced pluripotent stem cell (iPSC) differentiation by culture on frozen sections having a suitable microenvironment is possible. Induced PSCs were cultured on frozen sections of the liver, the brain, the spinal cord, and cover glasses (control) for 9 days. The iPSCs cultured on the sections of the liver resembled hepatocytes, whereas those on sections of the brain and the spinal cord resembled neuronal cells. The percentage of hepatocytic marker-positive cells in the iPSCs cultured on the sections of the liver was statistically higher than that of those in the iPSCs cultured on the sections of the brain and the spinal cord or on cover glasses. In contrast, the iPSCs cultured on the sections of the brain and the spinal cord revealed a high percentage of neural marker-positive cells. Thus, iPSCs can be differentiated into a specific cell lineage in response to specific factors within frozen sections of tissues/organs. Differentiation efficacy of the frozen sections markedly differed between the iPSC clones. Therefore, our induction method could be simple and effective for evaluating the iPSC quality.

## 1. Introduction

Research on previous medical technology has focused on the use of living and artificial organ transplantations rather than the use of tissues or organs that are either diseased or subjected to trauma, leading to complications such as rejection, durability, and mechanical strength. Recently, regenerative medicine using stem cells has garnered considerable attention because of its potential clinical applications. Regenerative medicine requires a stable source of stem cells, such as embryonic stem cells (ESCs) [1], adult stem cells (ADSCs) [2], and induced pluripotent stem cells (iPSCs) [3]. Each cell type has advantages and disadvantages related to its use in regenerative medicine. For example, ESCs and iPSCs are pluripotent, whereas ADSCs are multipotent [4]. Notably, there are ethical issues surrounding the use of ESCs in regenerative medicine [5] and compatibility problems with ESC transplantation [6]. Induced PSCs, which have pluripotency comparable with ESCs and are without ethical and social issues, are expected to be a stable source of stem cells that can be clinically employed in regenerative medicine.

The establishment of iPSCs has profoundly impacted both the basic biological knowledge and the field of clinical medicine. In 2006, iPSCs were first generated in mice by Takahashi and Yamanaka [3]. Subsequently, several groups have reported the successful generation of human iPSCs using similar strategies [7,8,9]. Induced PSCs are now generated using various reprograming methods from multiple somatic cell types, including dermal fibroblasts [10], adipocytic stem cells [11], and neural stem cells [12]. In dental medicine, iPSCs have been generated from dental pulp cells and dental ligaments [13]. The oral mucosa is considered to be a useful source for generating iPSCs as biopsies are not limited by the sex or age of the patient; the procedure is easy, minimally invasive, and repeatable, and the incision heals quickly and without scarring. Thus, we used iPSCs generated from human oral mucosa fibroblasts [14] to evaluate a novel method for differentiation and quality assessment.

There are various challenges to overcome to enable the use of iPSCs in regenerative medicine, including the establishment of a reliable and efficient method to induce differentiation. Numerous studies have reported methods for inducing differentiation of iPSCs into various functional cells, such as hematopoietic cells [15], vascular endothelial cells [16], myocardial cells [17], motor neurons [18], various neural cells [19], retinal pigment epithelial cells [20], photoreceptor cells [21], dendritic cells [22], macrophages [23], and pancreatic beta cells [24]. However, these methods depend upon appropriate environmental factors, such as culture media, substrates, and cytokines, and the determination of factors required for differentiation from the complicated milieu of the original tissue/organ is extremely difficult. Another issue that needs to be resolved is the establishment of a simple method for the assessment and management of iPSC quality. There is a well-known discrepancy between gene expression in and the differentiation potency of iPSCs. This discrepancy is affected by the original source of somatic cells for generation [25,26,27], reprograming method [26,27,28,29,30,31,32], generation laboratory [33], and extracellular matrix from feeder cells [34,35]. Additionally, advanced techniques are needed to maintain iPSCs in an undifferentiated state, which can be influenced by changes in cultivation conditions. Thus, development of a method for the assessment and management of iPSC quality and differentiation potency is urgently required to increase the use of these cells in regenerative medicine. We hypothesized that differentiation can be induced using frozen sections of the tissues/organs having a suitable microenvironment. Briefly, frozen sections were produced from the tissue/organ targeted for regeneration. Next, iPSCs were directly cultured on the frozen sections to induce differentiation through the direct action of growth and/or differentiation factors released from the sections. This method may also be used to select the optimal clone for the regeneration of a specific organ and assess the quality of iPSCs by comparing differentiated and undifferentiated clones.

## 2. Materials and Methods

This study includes data related to male ICR mice, it received an approval from the Animal Experiment Committee of Tsurumi University, approval number 26A024. All the methods were performed in accordance with the relevant guidelines and regulations.

### 2.1. Acute Liver Injury

Male 6-week-old ICR mice were used in these experiments. Acute liver injury was induced by an intraperitoneal injection of carbon tetrachloride at the dose of 1 mL/kg body weight (diluted 1:4 in olive oil). The mice were euthanized at 1, 2, 3, and 5 days after injection with an overdose of pentobarbital (intraperitoneal), and the livers were collected.

### 2.2. Preparation of Frozen Sections

Tissue samples included livers with drug-induced hepatitis, normal livers, spinal cords, and brains. The tissues were placed in a cryomold with an optimal cutting temperature (OCT) compound. The cryomold was placed in liquid nitrogen until the OCT compound froze (solid white color). The frozen sections were cut at a thickness of 5 μm with a cryostat and placed on cover glasses (Matsunami Glass Ind., Ltd., Tokyo, Japan). The cover glasses were washed twice with phosphate-buffered saline (PBS, pH 7.4) and dried at room temperature for 2 h.

### 2.3. Cell Culture and Differentiation of Human iPSCs

Induced PSCs were cultured on mitomycin C (Kyowa Hakko Kirin Co., Ltd., Tokyo, Japan)-treated SNL feeder cells and seeded on a type 1 collagen-coated dish (Iwaki Co. Ltd., Tokyo, Japan) in the hES medium (ReproCELL, Yokohama, Japan) supplemented with 4 ng/mL recombinant human basic fibroblast growth factor (Wako Pure Chemical Industries, Ltd., Osaka, Japan). The iPSCs were cultured under 5% CO_2_ in air at 37 °C, and the medium was changed every day. Induced PSC colonies were loosely detached by collagenase treatment and suspended in Dulbecco’s modified Eagle’s medium (Sigma-Aldrich Co, St. Louis, MO, USA) containing 5% fetal bovine serum. Suspended iPSC colonies were plated on the frozen sections in noncoated glass bottom dishes (Matsunami Glass Ind., Ltd., Tokyo, Japan). The iPSCs were cultured for 9 days, and the medium was changed every 3 days.

### 2.4. RT-PCR

Total RNA was extracted from the cells after 9 days in culture using TRIzol Reagent (Invitrogen Co, Carlsbad, CA, USA). Complementary DNA was generated from 1 μg of total RNA using a SUPERSCRIPT Pre-amplification System (Invitrogen Co, Carlsbad, CA, USA) according to the manufacturer’s protocols. RT-PCR was performed using a PCR Master Mix kit (Thermo Fisher Scientific, Waltham, MA, USA) with the following parameters: denaturation at 95 °C for 2 min, including denaturation at 95 °C for 25 s, annealing for 35 s, and extension at 72 °C for 65 s. The cycles were followed by final extension at 72 °C for 5 min. Conditions and primer sequences for PCR amplification are shown in Table 1. The GAPDH gene was used as an internal control for cDNA quantity and quality. PCR products were analyzed by means of ethidium bromide staining after separation by electrophoresis through a 2% agarose gel.

### 2.5. Immunocytochemistry

The cells cultured for 9 days were fixed with 2% paraformaldehyde in PBS (pH 7.4) for 60 min at room temperature. The fixed cells were washed two times with PBS and permeabilized with TRITON X-100 (ICN Biomedical, Irvine, CA, USA) in PBS containing 0.1% BSA for 15 min and then blocked with 10% normal serum from the same animal species as the secondary antibody for 10 min. Next, the samples were incubated overnight at 4 °C with rabbit anti-AFP antibody (Abcam, Cambridge, UK) diluted 1:500, goat anti-α1-antitrypsin antibody (Abcam) diluted 1:300, rabbit anti-glial fibrillary acidic protein antibody (Sigma-Aldrich Co, St. Louis, MO, USA) diluted 1:80, or rabbit anti-CNPase antibody (Abcam) diluted 1:100 in PBS containing 1% BSA. After washing three times with PBS, the samples were incubated with FITC-conjugated anti-goat IgG (Wako Pure Chemical Industries, Ltd., Osaka, Japan) and Alexa 594-conjugated anti-rabbit IgG (Invitrogen Co, Carlsbad, CA, USA) as secondary antibodies for 1 h. The nuclei were counterstained with 4′,6-diamidino-2-phenylindole. Then, the cells were analyzed using fluorescence microscopy.

### 2.6. Differentiation-Inducing Effects

The cells cultured on the frozen sections were stained with hepatocyte- and neuron-specific markers. One hundred fields (10 sections × 10 fields) of interest (730 × 500 μm each) for each frozen section were randomly selected, and the number of AFP-positive, AAT-positive, GFAP-positive, and CNPase-positive cells was counted. The percentage of positively stained cells out of the nucleated cell population was calculated for each frozen section. The data were analyzed using Scheffe’s test; *p* < 0.05 was considered statistically significant (SPSS, Inc. Chicago, IL, USA).

## 3. Results

### 3.1. Differentiation of iPSCs Cultured on Frozen Sections of the Liver

We used iPSCs generated from the oral mucosa by means of retroviral four-factor gene transfer (termed hOF-iPSCs) (see Appendix A). We prepared frozen sections of mouse livers because hOF-iPSCs are easy to culture on sections of this organ. We also attempted to culture hOF-iPSCs on frozen sections of the livers with drug-induced hepatitis because we hypothesized that the liver’s enhanced regenerative capability upon damage induced by hepatitis provides an advantageous environment for inducing hepatocyte differentiation (see Appendix A). We examined differentiation of the hOF-iPSCs cultured on frozen sections of the normal livers (normal liver group) and the livers with hepatitis (hepatitis liver group) (see Appendix A); hOF-iPSCs were seeded on the frozen sections and cultured for 9 days, after which hOF-iPSCs appeared to be loose and spread. The hOF-iPSCs cultured on cover glasses exhibited various morphological changes; however, the hOF-iPSCs cultured on frozen sections of the normal livers and the livers with hepatitis exhibited large and polygonal morphological changes resembling hepatocytes (Figure 1a). To better characterize the differentiated hOF-iPSCs, the gene expressions of hepatocyte-differentiated markers, including Sox17, forkhead box protein A2 (FOXA2), α-fetoprotein (AFP), α1-antitrypsin (AAT), albumin (ALB), and cytochrome P450 3A4 (CyP3A4); neuron-differentiated markers, including nestin, myelin basic protein (MBP), 2′,3′-cyclic nucleotide 3′-phosphodiesterase (CNPase), glial fibrillary acid protein (GFAP), neurofilament 200 (NF200), and β-tubulin 3 (Tuj1) as the control; and undifferentiated cell markers (Nanog and Oct3/4) were examined by reverse transcription polymerase chain reaction (RT-PCR). The control hOF-iPSCs demonstrated high levels of undifferentiated cell markers with a low level of hepatocyte- and neuron-differentiated markers. In contrast, induced hOF-iPSCs in the normal and hepatitis liver groups exhibited hepatocyte-differentiated markers but not undifferentiated cell markers or neuron-differentiated markers (see Appendix A). Next, we examined the protein expression of hepatocyte-differentiated markers (AFP and AAT) and neuron-differentiated markers (GFAP and CNPase). Compared with the control hOF-iPSCs, the hOF-iPSCs cultured on the normal livers and the livers with hepatitis significantly expressed AFP and AAT (Figure 1b). Neuron-differentiated markers were minimally expressed in all the hOF-iPSCs groups (Figure 1b). Fields (730 × 500 µm/field, 10 fields) were randomly selected, and the number of AFP- and AAT-positive hOF-iPSCs was counted. The percentages of AFP- and AAT-positive hOF-iPSCs in the normal and hepatitis liver groups were significantly higher than those in the control cells (Figure 1c). There was no significant difference between the normal liver group and the day 1 hepatitis liver group. The percentages of AFP- and AAT-positive differentiated hOF-iPSCs decreased in the hepatitis liver group on days 2, 3, and 5 compared with those on day 1.

These data confirm that the hOF-iPSCs differentiated into hepatocytes after culturing on frozen sections of the liver. Notably, there was no difference between the use of frozen sections of the livers with or without hepatitis. To expand the applicability of this method, we induced differentiation of the hOF-iPSCs using frozen sections of other tissues.

### 3.2. Differentiation of hOF-iPCS Cultured on Frozen Sections of the Brain and the Spinal Cord

Next, we attempted to induce differentiation of hOF-iPSCs to neural cells by culturing on frozen sections of the brain and the spinal cord (see Appendix A); hOF-iPSCs were seeded on frozen sections of the brain and the spinal cord and cultured for 9 days. Subsequently, all the hOF-iPSCs appeared to be loosely spread. The hOF-iPSCs cultured on cover glasses as the controls revealed various morphological changes, whereas the hOF-iPSCs cultured on frozen sections of the brain and the spinal cord developed neuronal morphological traits. The cytoplasm in flat hOF-iPSCs was retracted toward the nucleus, forming a contracted multipolar cell body with membranous, process-like peripheral extensions. The cell bodies became increasingly spherical and refractile, exhibiting a typical neuronal perikaryal appearance (Figure 2a). To further characterize these differentiated hOF-iPSCs, the gene expression of hepatocyte-differentiated markers (*SOX17*, *FOXA2*, *AFP*, *AAT*, *ALB*, and *CYP3A4*), neuron-differentiated markers (*NES*, *MBP*, *CNP*, *GFAP*, *NF200*, and *TUJ1*), and undifferentiated cell markers *(NANOG* and *OCT3*/*4*) was determined using RT-PCR (see Appendix A). The control hOF-iPSCs mainly expressed undifferentiated cell markers, with a minority of cells exhibiting hepatocyte- and neuron-differentiated markers. In contrast, the hOF-iPSCs cultured on sections of the brain and the spinal cord expressed neuron-differentiated markers, whereas undifferentiated cell and hepatocyte-differentiated markers were not expressed (see Appendix A). We also assessed the protein expression of hepatocyte-differentiated markers (AFP and AAT) and neuron-differentiated markers (GFAP and CNPase) using immunocytochemistry. The hOF-iPSCs cultured on sections of the brain and the spinal cord revealed marked GFAP and CNPase expression compared with the control hOF-iPSCs (Figure 2b). In contrast, minimal expression of hepatocyte-differentiated markers was observed (Figure 2b). Fields of interest (730 × 500 µm/field, 10 fields) were randomly selected, and the number of GFAP-positive and CNPase-positive hOF-iPSCs was counted. The number of GFAP-positive or CNPase-positive hOF-iPSCs cultured on sections of the brain and the spinal cord was significantly higher than that in the control group (Figure 2c). No significant difference was observed between sections of the brain and the spinal cord.

Thus, hOF-iPSCs can be induced to be differentiated into neural cells by culturing on frozen sections of the brain and the spinal cord. To highlight the applications of our culture method, we next attempted to induce differentiation of iPSCs generated from different tissue sources or by means of different reprograming methods.

### 3.3. Differentiation of iPSCs Cultured on Frozen Sections

Induced PSCs, HPS63 (generated from human skin fibroblasts by means of retroviral four-factor gene transfer) [7], HPS76 (generated from human skin fibroblasts by means of episomal six-factor gene transfer) [13], and HPS77 (generated from human dental pulp cells by means of episomal six-factor gene transfer) [13] were obtained from the RIKEN Cell Bank to confirm that frozen sections can induce differentiation of these cell types (see Appendix A). The iPSCs cultured on frozen sections of the liver resembled hepatocytes, whereas those cultured on sections of the brain and the spinal cord resembled neural cells (see Appendix A). The protein expression of hepatocyte-differentiated markers (AFP and AAT) and neuron-differentiated markers (GFAP and CNPase) was examined using immunocytochemistry. The iPSCs cultured on sections of the liver revealed higher AFP and AAT expression than the control group or the iPSCs cultured on sections of the brain and the spinal cord. Likewise, the iPSCs cultured on sections of the brain and the spinal cord exhibited higher GFAP and CNPase expression than the control group or the liver group (see Appendix A). Fields of interest (730 × 500 µm/field, 10 fields) were randomly selected, and the number of AFP-, AAT-, GFAP-, and CNPase-positive iPSCs was counted. The iPSCs cultured on sections of the liver exhibited a high percentage of AFP- and AAT-positive staining, whereas the iPSCs cultured on sections of the brain and the spinal cord revealed a high percentage of GFAP- and CNPase-positive staining (Figure 3).

These results are summarized in Table 1. Induced PSCs generated from somatic cells by means of different reprograming methods were induced to differentiate into hepatocyte-like or neuron-like cells by culturing on frozen sections of the liver or of the brain and the spinal cord. This novel and simple method of inducing differentiation using frozen sections of the target tissue is reproducible and applicable to regenerative medicine involving iPSCs. Moreover, specific iPSCs can be screened and selected for differentiation using this method; thus, the quality of iPSCs can be easily assessed and managed.

### 3.4. Mechanism of iPSCs Cultured on Frozen Sections

The HPS77 cells cultured on frozen sections of the liver, the brain, and the spinal cord were fixed with 4% PFA or cold acetone. The HPS77 cells cultured on sections of the non-fixed normal liver exhibited higher AFP expression than those on the liver sections fixed with 4% PFA or cold acetone. Additionally, the HPS77 cells cultured on sections of the non-fixed normal brain and the spinal cord exhibited higher GFAP and CNPase expression than those on the brain and spinal cord sections fixed with 4% PFA or cold acetone. Meanwhile, the AAT expression varied significantly between the normal frozen sections and in the frozen sections fixed with 4% PFA or cold acetone (Figure 4).

## 4. Discussion

Our research provides a novel, reliable, and efficient method for inducing differentiation of iPSCs using frozen sections of tissues/organs. Here, we induced differentiation of the hOF-iPSCs cultured on cover glasses as the control and frozen sections of the normal livers and the livers with drug-induced hepatitis and sections of the brain and the spinal cord. The hOF-iPSCs cultured on the cover glass exhibited various morphological changes, and these hOF-iPSCs expressed several markers in disorder at the gene level. However, the hOF-iPSCs cultured on sections of the normal livers or the livers with hepatitis exhibited a relatively large and polygonal morphological shape that resembled hepatocytes, and these hOF-iPSCs expressed hepatocyte-differentiated markers at the gene and protein levels. Similarly, the hOF-iPSCs cultured on sections of the brain and the spinal cord developed neuronal morphological traits and expressed neuron-differentiated gene and protein markers. These data confirm that frozen sections of the target tissues can be easily used to induce differentiation of hOF-iPSCs. The hOF-iPSCs cultured on sections of the livers with hepatitis, which were considered to have enhanced original regenerative capability caused by damage, did not exhibit any significant difference in the percentage of AFP- and AAT-positive staining compared with the cells cultured on sections of the normal livers. In addition, compared with the control group, the number of AFP- and AAT-positive hOF-iPSCs cultured on sections of the livers with hepatitis decreased on days 2, 3, and 5. This phenomenon could be due to hepatitis-mediated changes in the extracellular matrix and signaling molecules, e.g., inflammatory cytokines, which adversely affected differentiation.

We also demonstrated that our method could be used to induce differentiation of iPSCs from various sources and by means of different reprograming methods. We induced differentiation of iPSCs (HPS63, HPS76, and HPS77) into hepatocyte-like cells by culturing them on frozen sections of the liver or into neuron-like cells by culturing them on frozen sections of the brain and the spinal cord. Each iPSC type exhibited a high differentiation rate. Thus, our new method using frozen sections of the target tissues/organs is reproducible and applicable in regenerative medicine involving iPSCs.

Notably, we observed a difference in the efficiency of induction between the iPSC clones. For example, HPS63 favored differentiation into hepatocyte-like cells compared with other iPSCs, whereas HPS76, which were generated from skin, favored differentiation into neuron-like cells compared with HPS63. HPS77 also favored differentiation into neuron-like cells. In contrast, hOF-iPSCs were readily induced to differentiate into hepatocyte-like cells and neuron-like cells compared with the other iPSCs used in this study. These differences may be reflected in the reactivity of the specific iPSCs to induced differentiation. However, additional frozen sections of tissues/organs and iPSCs generated from different sources and by means of different reprograming methods are required to test this hypothesis.

Until now, several studies have been reported to evaluate and overcome the variation of differentiation potency between iPSCs, which is affected by the original source of somatic cells, reprogramming method, and donor difference [26,36,37,38]. Compared to these studies, our method is simple to assess and manage the quality of the iPSCs to be used in regenerative medicine. Our method is a new challenge to induce the differentiation ability of iPSCs by only seeding on the frozen section, without adding any proteins, chemicals, or exogenous gene transfer. Furthermore, the methods adding the known differentiation-inducing factors have not been able to sufficiently induce the differentiation of iPSCs so far. It may be possible to understand the missing molecular mechanism of cellular differentiation using our method.

Moreover, we considered the differentiation-inducing mechanism of our method as follows.

Firstly, we attempted to induce differentiation of human iPSCs on mouse tissues because it is realistically difficult to obtain human tissue, and many cases of in vitro studies have been conducted using mouse tissues and applied to human tissues. Therefore, we challenged whether human iPSCs can be affected by mouse-derived materials. The reason why mouse tissues can induce differentiation for human iPSCs is still unknown. The similarity of protein sequence or microstructure of our examined tissues between mice and humans might be considered, but detailed studies need to be conducted in the future.

Secondly, if our method is to use heterologous animal tissues, it is necessary to further investigate the factors that induce differentiation to use the clinical applications in the future. The protein, cellular ultrastructure, and micro interference RNA (miRNA) are considered as the factors that contribute to the differentiation induced by our method. For example, the reason for considering influence of the protein is that we observed efficiency of induction decrement in the frozen sections fixed with 4% PFA or cold acetone. Thus, the result that the differentiation efficiency was decreased by fixed processing suggests that the biologically active protein was one of the elements important to inducing differentiation. Moreover, the result that the differentiation efficiency remained higher in the control samples than in the fixed processing group suggested that the mechanism of our method was composed of a biologically active protein and several elements besides the activated protein lost by fixed processing. The aspect in which cellular ultrastructure and miRNA of frozen sections have a differentiation-led ability was suggested. Because our method consisted in the induction of differentiation based on the environment associated with the organ, the organ revealed a possibility of constituting the differentiation-inducing property of a stem cell. So far, preliminary examinations of cellular ultrastructure and miRNAs have been carried out, but it has not been clarified yet.

Altogether, our results strongly suggest that using frozen sections of tissues to culture iPSCs is a new and simple method for inducing differentiation and assessing and managing the quality of the iPSCs to be used in regenerative medicine.

## 5. Conclusions

In this study, we clearly demonstrated that iPSCs are capable of differentiating into a specific cell lineage in response to certain factors contained in frozen sections of tissues/organs. Interestingly, the differentiation efficiency of iPSCs on frozen sections was clearly different between the iPSC clones. Judging from these facts, this induction method for human iPSCs using frozen sections is considered to be useful as a simple and effective means for inducing differentiation of iPSCs and also for evaluating the potential and/or quality of iPSCs.

## Figures and Tables

**Figure 1 cells-10-02827-f001:**
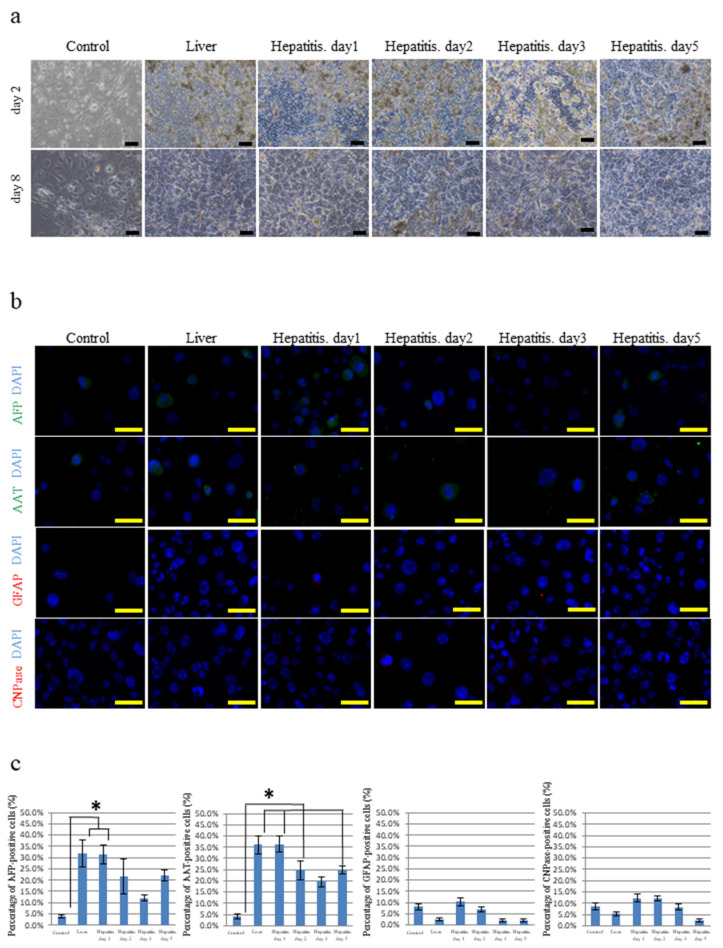
Analysis of a new method for inducing differentiation of iPSCs generated from the oral mucosa (hOF-iPSCs) by culturing on frozen sections of normal livers and livers with hepatitis. (**a**) Phase-contrast microscopy of the hOF-iPSCs cultured on cover glasses (as the controls) and frozen sections of normal livers or livers with hepatitis. On day 2, all the hOF-iPSCs cultured on the cover glasses and the frozen sections were loosely spread. On day 8, the hOF-iPSCs cultured on the cover glasses exhibited various morphological changes, whereas the hOF-iPSCs cultured on the frozen sections of both the normal livers and the livers with hepatitis exhibited large and polygonal morphological changes resembling hepatocytes. (**b**) Immunocytochemical analysis of hepatocyte-differentiated (AFP and AAT) and neuron-differentiated (GFAP and CNPase) markers in induced hOF-iPSCs. As a counterstain, 4′,6-diamidino-2-phenylindole (DAPI) was used. AFP and AAT were efficiently expressed in the induced hOF-iPSCs cultured on the normal livers and the livers with hepatitis compared with the control hOF-iPSCs. Few neuron-differentiated markers were expressed in the hOF-iPSCs. (**c**) The percentages of AFP-, AAT-, GFAP-, and CNPase-positive hOF-iPSCs in the selected fields of interest. The hOF-iPSCs cultured on frozen sections of both types of livers exhibited higher AFP- and AAT-positive staining than the control group. No significant differences were observed between the normal liver group and the hepatitis liver group. Scale bars: 100 µm; * *p* < 0.05.

**Figure 2 cells-10-02827-f002:**
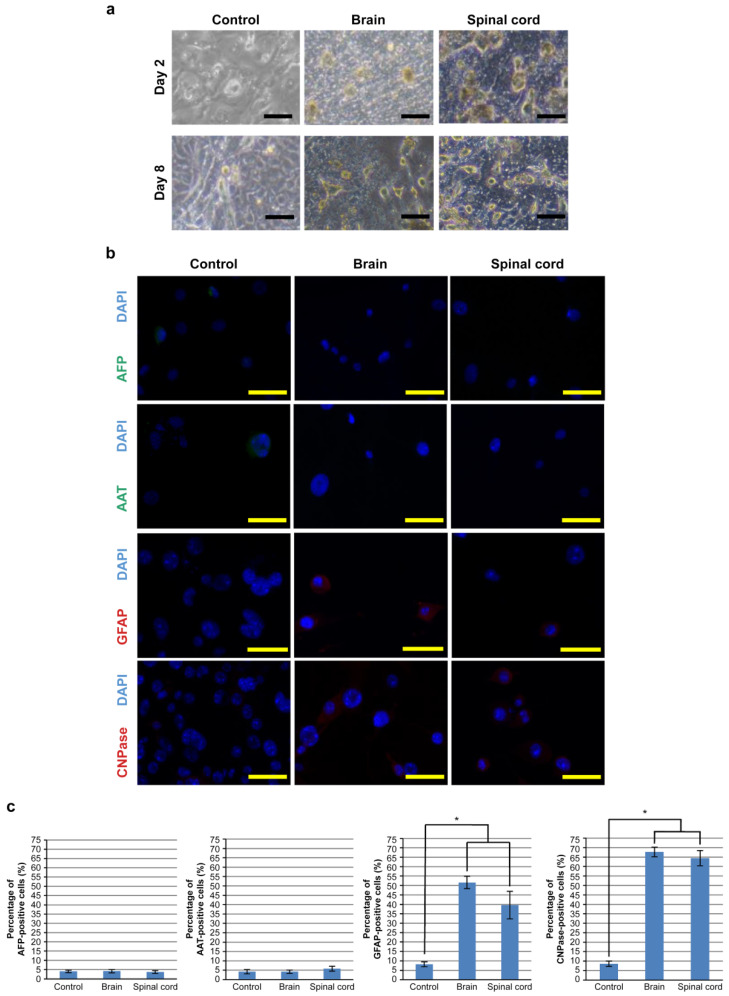
Analysis of the method for inducing differentiation of hOF-iPSCs cultured on frozen sections of the brain and the spinal cord. (**a**) Phase-contrast microscopy of the hOF-iPSCs cultured on cover glasses (as the controls) and frozen sections of the brain and the spinal cord. On day 2, all the hOF-iPSCs cultured on the cover glasses or the frozen sections were loosely spread. On day 8, the hOF-iPSCs cultured on the cover glasses exhibited various morphological changes, whereas the hOF-iPSCs cultured on frozen sections of the brain and the spinal cord exhibited neuronal morphological traits. The cytoplasm in flat hOF-iPSCs was retracted toward the nucleus, forming a contracted multipolar cell body and membranous, process-like peripheral extensions. Cell bodies became increasingly spherical and refractile, exhibiting a typical neuronal perikaryal appearance. This tendency was observed in the cells from both sections of the brain and spinal cord. (**b**) Immunocytochemical analysis of hepatocyte-differentiated (AFP and AAT) and neuron-differentiated (GFAP and CNPase) markers in the induced hOF-iPSCs. DAPI was used as a counterstain. The hOF-iPSCs cultured on sections of the brain and the spinal cord exhibited higher GFAP and CNPase expression than the control group. In contrast, very few hepatocyte-differentiated markers were observed. (**c**) The percentages of AFP-, AAT-, GFAP-, and CNPase-positive hOF-iPSCs in the selected fields of interest. The hOF-iPSCs cultured on sections of the brain and the spinal cord exhibited higher GFAP- and CNPase-positive staining than the control group. No significant differences were observed between sections of the brain and the spinal cord. Scale bars: 100 µm; * *p* < 0.05.

**Figure 3 cells-10-02827-f003:**
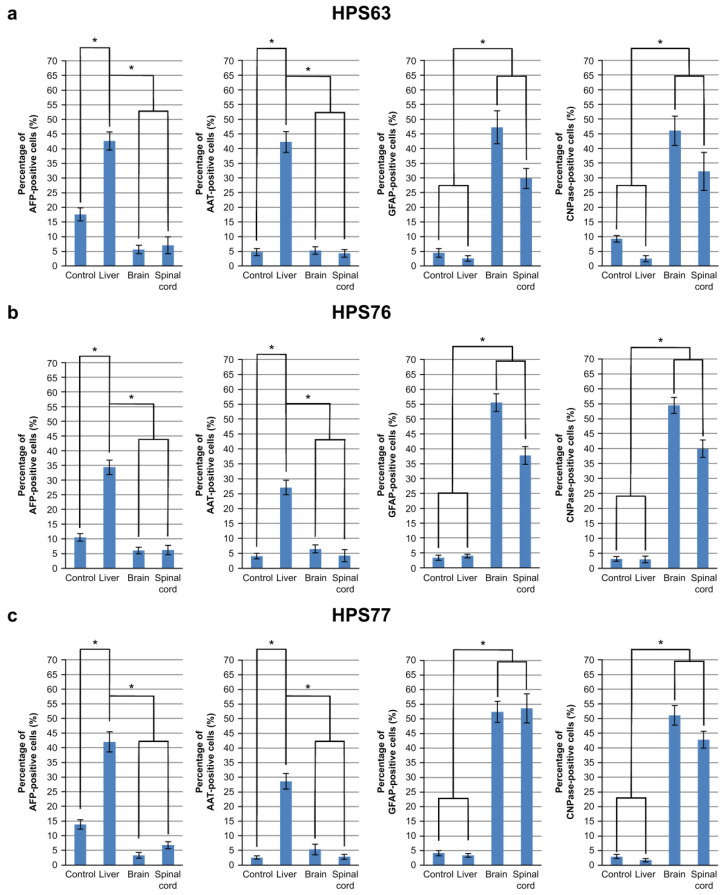
Protein expression of cell markers in differentiated iPSCs. The percentages of AFP-, AAT-, GFAP-, and CNPase-positive iPSCs in the selected fields of interest. (**a**) The HPS63 cultured on frozen sections of the liver exhibited significantly more AFP- and AAT-positive cells than the control group and the HPS63 cultured on sections of the brain and the spinal cord. The HPS63 cultured on sections of the brain and the spinal cord exhibited significantly more GFAP- and CNPase-positive cells than the control group and the HPS63 cultured on sections of the liver. (**b**) HPS76 exhibited a pattern similar to hOF-iPSCs and HPS63. (**c**) HPS77 also exhibited a pattern similar to hOF-iPSCs, HPS63, and HPS76. Thus, all the iPSCs exhibited similar expression patterns. Note: * *p* < 0.05.

**Figure 4 cells-10-02827-f004:**
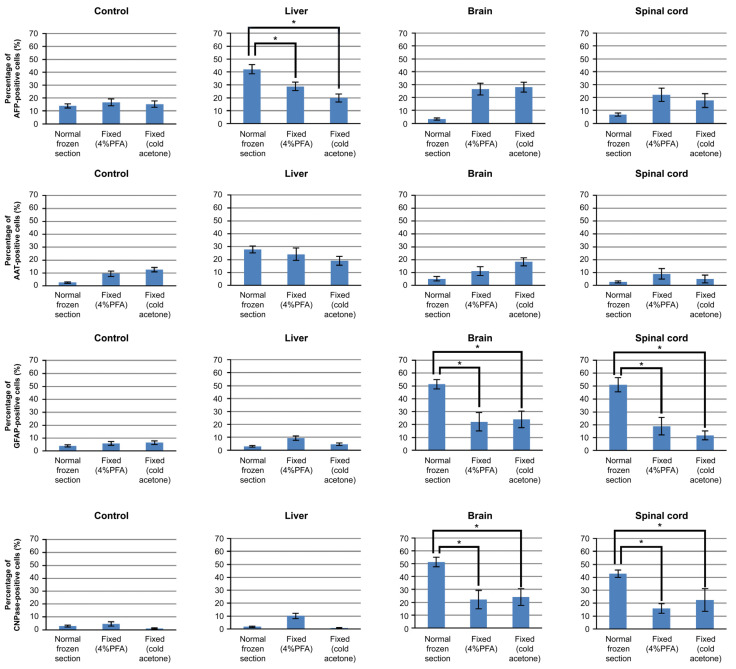
Immunocytochemical analysis for the expression of hepatocytic (AFP and AAT) and neurogenic (GFAP and CNPase) markers in the HPS77 cells cultured on the unfixed or fixed frozen sections with 4% PFA or cold acetone. Statistically more HPS77 cells expressed hepatocytic markers (AFP) when cultured on unfixed frozen sections of the liver compared with when cultured on the fixed frozen sections. On the contrary, more HPS77 cells expressed neural markers (GFAP and CNPase) when cultured on unfixed frozen sections of the brain/spinal cord than when cultured on the fixed frozen sections. Note: * *p* < 0.05.

**Table 1 cells-10-02827-t001:** Characteristics of the iPSCs used and their efficiency of induction.

		hOF-iPSCs	HPS63	HPS76	HPS77
Origin		oral mucosa	skin	skin	dental pulp
Gene expression		retrovirus vector	retrovirus vector	episomal vector	episomal vector
Foreign gene		*OCT3/4*, *SOX2*, *KLF4*, *C-MYC*	*OCT3/4*, *SOX2*, *KLF4*, *C-MYC*	*OCT3/4*, *SOX2*, *KLF4*, *L-MYC*, *LIN28*, *P53 SHRNA*	*OCT3/4*, *SOX2*, *KLF4*, *L-MYC*, *LIN28*, *P53 SHRNA*
AFP	Control	4.03%	17.59%	10.53%	13.82%
	Liver	31.88%	42.67%	34.35%	42.01%
	Brain	4.15%	5.65%	6.07%	3.31%
	Spinal cord	3.70%	7.07%	6.25%	6.80%
AAT	Control	4.19%	4.72%	4.09%	2.51%
	Liver	36.19%	42.20%	27.07%	27.82%
	Brain	4.24%	5.25%	6.47%	5.17%
	Spinal cord	5.80%	4.32%	4.22%	2.69%
GFAP	Control	8.21%	4.48%	3.34%	4.00%
	Liver	5.37%	2.57%	3.99%	3.03%
	Brain	51.64%	47.24%	55.51%	51.30%
	Spinal cord	39.57%	29.83%	37.75%	51.01%
CNPase	Control	8.59%	9.25%	3.10%	2.89%
	Liver	5.37%	2.51%	2.89%	1.62%
	Brain	67.66%	46.02%	54.39%	51.63%
	Spinal cord	64.39%	32.18%	39.95%	42.72%

## Data Availability

The data presented in this study are available in the Appendix A.

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
