# Peer review of "A New Induction Method for the Controlled Differentiation of Human-Induced Pluripotent Stem Cells Using Frozen Sections"

_cells, 2021, doi:10.3390/cells10112827_

Round 1

Reviewer 1 Report

  1. What is the magnification of the fluorescence images in each figure? Can the authors add an overall image of the differentiated cells with lower magnification?
  2. The term AFP-, AAT-, GFAP-, and CNPase-positive iPSCs is confusing. Aren’t these cells already thought to be differentiated into a specific lineage? It might be better to change this term.
  3. The control cells morphology does not look like iPSCs, did the authors coat the cover glasses of the control group with the coating reagents for iPSC culture, such as Matrigel or vitrogentin? What matrix were the iPSCs maintained?
  4. Also, in figure S1, the authors said that the iPSCs expressed the undifferentiated markers, however, the data is not shown.
  5. Did the authors confirm the protein expression of the specific markers for each lineage?
  6. Is there any other references that might be related to this study?
  7. The authors should add the headquarters address in the material and method section.

Author Response

Dr. Reviewer #1

Re:   Manuscript ID: cells-1417824

Title: A new induction method for the controlled differentiation of human-induced pluripotent stem cells using frozen sections

     Thank you very much for your decision letter of Oct 7, 2021 including your comments concerning our manuscript entitled “A new induction method for the controlled differentiation of human-induced pluripotent stem cells using frozen sections.” I’m returning herewith the revised manuscript again.

     We have carefully studied your comments and have made necessary corrections, which is indicated by colored text in the revised manuscripts. In addition to your comments, we also revised some other parts in the text including figure legends, and figures, by colored text to improve.

     Our response to your comments is as attached file:

We believe the manuscript has been improved satisfactorily and hope that it is now acceptable for publication in Cells. 

Sincerely,

Susumu Tadokoro, DDS, PhD

Reviewer 2 Report

Tadokoro et al, in this manuscript showed a method to differentiate human induced pluripotent stem cells (hiPSCs) by growing them on frozen section of mouse tissues (liver, brain, and spinal cord). Authors claimed that these tissue sections provide microenvironment and growth supplement to differentiate hiPSCs into specific cell types. Authors characterized these cell types by analyzing cell specific markers by gene expression analysis and immunostaining.  Authors used different hiPSC lines and showed that the differentiation potential of tissue specific frozen section is reproducible with different hiPSCs lines. Finally, authors demonstrated that this differentiation potential is reduced on fixed tissue sections compared to fresh frozen section.  Based on these data authors concluded that frozen section of tissues to culture hiPSCs and differentiate them into specific cell types. Authors claimed that, this is simple method of differentiation and utilizes to access the quality of hiPSC lines for regenerative medicine. This is an important study and tissue, or organ specific microenvironment and growth factors provides appropriate condition to differentiate hiPSCs into specific cell type. 
This is study raises following concerns and this manuscript can be further improved by addressing the major following concerns.
1. Differentiating hiPSCs into specific cell type using frozen mouse tissue/ organ section does not provide the defined conditioned and it will be highly variable based on quality of tissue/ organ sections. There are well defined methods to differentiated hiPSCs into hepatocytes (Kajiwara, M. et al 2012 and Peters, Henderson and Warren et al 2016 etc), hiPSCs into different neuronal cell types (Hu et al 2010 and Mahajani et al 2019 etc). How authors will address these challenges.
2. How authors will explain that mouse tissue/ organ microenvironment and growth factors will efficiently support differentiation of human iPSCs into respective cell types.
3. It would be more informative if authors showed which growth factors secreted by frozen sections and what extracellular matrixes present in the frozen section, which support differentiation, will provide very important information and can be utilize to develop methods to efficiently differentiate these cell types under defined, animal component free conditions and ultimately used for clinical applications.

Author Response

Dr. Reviewer #2

Re:   Manuscript ID: cells-1417824

Title: A new induction method for the controlled differentiation of human-induced pluripotent stem cells using frozen sections

     Thank you very much for your decision letter of Oct 7, 2021 including your comments concerning our manuscript entitled “A new induction method for the controlled differentiation of human-induced pluripotent stem cells using frozen sections.” I’m returning herewith the revised manuscript again.

     We have carefully studied your comments and have made necessary corrections, which is indicated by colored text in the revised manuscripts. In addition to your comments, we also revised some other parts in the text including figure legends, and figures, by colored text to improve.

     Our response to your comments is as attached file:

We believe the manuscript has been improved satisfactorily and hope that it is now acceptable for publication in Cells. 

Sincerely,

Susumu Tadokoro, DDS, PhD

Round 2

Reviewer 2 Report

In the revised manuscript authors addressed comments and suggestions made by the reviewers.